# Cancer symptom clusters in adult patients undergoing chemotherapy: A systematic review and meta-analysis protocol

**Luís Carlos Lopes-Júnior**[1,2]*, **Taiani Ferrarini**[1], **Luciana Bicalho Cevolani Pires**[2], **Jonathan Grassi Rodrigues**[3], **Luciane Bresciani Salaroli**[2], **Karolini Zuqui Nunes**[1]

**1** Health Sciences Center at the Federal University of Espírito Santo (UFES), Vitoria, Espírito Santo, Brazil, **2** Graduate Program in Nutrition and Health at the Federal University of Espírito Santo (UFES), Vitoria, Espírito Santo, Brazil, **3** Graduate Program in Public Health at the Federal University of Espírito Santo (UFES), Vitoria, Espírito Santo, Brazil

\* lopesjr.lc@gmail.com

## Abstract

### Background

In oncology, the prevalence of symptoms is preferentially analyzed in isolation instead of being considered in clusters. However, clinical practice shows that symptoms rarely occur separately but rather form clusters that share common underlying mechanisms in terms of intensity and severity, creating a synergistic effect among them, which can even help predict the development of future symptoms.

### Objective

To identify and gather evidence on the prevalence, composition, severity, and predictors of cancer symptom clusters in adult cancer patients undergoing chemotherapy.

### Methods

This systematic review and meta-analysis protocol was developed in compliance with PRISMA-P. Observational and experimental study designs will be included. MEDLINE/PubMed, Cochrane Library, Embase, CINAHL, PsycINFO, Web of Science, Scopus, LILACS, clinical trials.gov-NIH, the British Library, Google Scholar, and preprints [medRXiv] will be searched with no restrictions on idioms, dates, or settings. Two investigators will independently select the studies, perform data extraction, and critically appraise the risk of bias of the included studies. Heterogeneity among the studies will be assessed using the $I^2$ statistic. If meta-analysis was feasible, a random-effect model analysis will be carried out. For data analysis, the pooled effect will be estimated considering 95% confidence interval and $\alpha$ = 5%. In addition, the certainty of evidence will be rated based on Cochrane methods in accordance with the Grading of Recommendations, Assessment, Development, and Evaluation (GRADE).

**Data Availability Statement:** Lopes-Júnior LC, Nunes KZ, Ferrarini T. Cancer symptom clusters in adult patients undergoing chemotherapy: a

systematic review and metanalysis protocol. 2022. Open Science Framework Repository. https://doi.org/10.17605/OSF.IO/K2GPT.

**Funding:** This research was carried out with the support of the Fundação de Amparo à Pesquisa e Inovação do Espírito Santo (FAPES) – "FAPES Notice No. 03/2021 – (Universal) – Process Number: 432/2021".

**Competing interests:** The authors have declared that no competing interests exist.

## Discussion

To the best of our knowledge, this systematic review and meta-analysis will be the first to identify and critically assess evidence regarding the prevalence, composition, severity, and predictors of cancer symptom clusters in adult cancer patients undergoing chemotherapy. We intend to provide health professionals with subsidies to reflect on a better understanding of symptom clusters in adult cancer patients, with the aim of contributing to the development of evidence-based therapeutic interventions and success in clinical practice.

## PROSPERO registration number

CRD42021248406.

## Introduction

Tumor progression and antineoplastic therapy can result in the manifestation of numerous symptoms during and after treatment, including cancer-related fatigue (CRF), sleep disturbances, pain, cognitive dysfunction, and anxiety [1, 2]. These symptoms reduce an individual's functional status, negatively impact quality of life, and may occur alone or together, constituting clusters of symptoms [1–4]. A cluster of symptoms was initially identified in experiments with mice subjected to the induction of infectious conditions (via administration of bacterial products such as LPS11) and administration of pro-inflammatory cytokines (interleukin [IL]-1β, IL-6, tumor necrosis factor [TNF]-α) [5], exhibiting a phenomenon called sickness behavior [6, 7].

The term sickness behavior refers to a set of behavioral changes that progress with diverse infectious and inflammatory processes, for instance, the apparent loss of interest in daily activities, such as searching for food, social interaction, and sex [6]. Studies in animals and humans have shown that cytokine infusion (either systemic or central) also induces sickness behavior [8]. Similarly, in cancer patients, a phenomenon of sickness behavior is expressed, such as cancer pain, CRF, cognitive and sleep disorders, depressed mood, anxiety and depression associated with high levels of pro-inflammatory cytokines expression, including IL-1β, IL-6, IL-8, TNF-α, IL-12p70, and interferon-gamma (IFN-γ) [5–8].

There is growing consistency that common biological mechanisms may underlie the interaction between the nervous, endocrine, and immune systems, which orchestrate a set of responses capable of inducing behavioral and physiological changes in cancer patients [5–11]. In particular, studies addressing sickness behavior as well as cancer symptom clusters in cancer patients support the hypothesis that pro-inflammatory cytokines are related to the biological mechanisms underlying the emergence of these clusters [1, 5, 7, 10, 12, 13]. Evidence suggests a strong association between depression, anxiety, cachexia, and high levels of cytokines expression (IL-1β, IL-6, IL-10, TNF-α, INF-γ, and fractalkine [CX3X]) in cancer patients [14].

Studies have shown that alterations in genes encoding pro-inflammatory cytokines (IL-1β, IL-6) and their high concentrations greatly contribute to the occurrence, intensity, and severity of various symptoms in cancer patients [15, 16]. A study conducted on 599 patients recently diagnosed with lung cancer revealed that an additive effect of the mutant alleles of IL-1β, IL-10, and TNFR2 was predictive of severe pain, depressed mood, and CRF in these patients [17].

It should be noted that oncology research preferentially focuses on the prevalence of symptoms analyzed in isolation, instead of considering them as clusters. However, clinical practice

shows that symptoms rarely occur separately but rather form clusters that share common underlying mechanisms in terms of intensity and severity, creating a synergistic effect among them, and can even predict the development of future symptoms [1, 2, 4, 5, 9, 10, 12, 14].

A previous systematic review of observational studies aimed to systematically assess the composition, longitudinal stability, and consistency across methodologies of common symptom clusters and their common predictors; however, it limited the search to patients with advanced cancer, focusing especially on the various statistical methods used. In addition, the authors used a generic tool for methodological assessment rather than specific tools for observational design to assess the risk of bias in the included studies [18].

Recently, a systematic review was published with the objective of evaluating the progress in symptom cluster research in adults receiving primary or adjuvant chemotherapy since 2016 and showed that psychological, gastrointestinal, and nutritional clusters were the most commonly identified clusters. Only the psychological clusters remained relatively stable over time [19].

Our review study differs in the following aspects: we did not specified a publication date or language limit (in order to minimize publication bias). Furthermore, we explored beyond observational studies by including experimental studies, and expanded the number of databases to eight, in addition to accessing gray literature and pre-prints for Health Sciences. In addition, both previous reviews used only generic tools for methodological appraisal and assessed the report of the study more than the risk of bias (in terms of internal and external validity of the studies). We used valid and design-specific tools following the recommendations of the Cochrane Collaboration [20]. Additionally, there are no systematic reviews or meta-analyses that considers a robust assessment of the risk of bias in studies involving cancer symptom clusters with validated and design-specific tools, which justifies the potential contribution of our study to the area.

This systematic review and meta-analysis aimed to identify and gather evidence on the prevalence, composition, severity, and predictors of cancer symptom clusters in adult cancer patients undergoing chemotherapy.

## Materials and methods

This systematic review and meta-analysis is in compliance with the Preferred Reporting Items for Systematic Reviews and Meta-Analyses Protocols (PRISMA-P) [21]. In addition, registration was obtained using PROSPERO/UK (registration ID: CRD42021248406).

### Search strategy

Eight electronic databases will be searched, including MEDLINE/PubMed, Cochrane Library, Embase, Web of Science, CINAHL, Scopus, PsycINFO, and LILACS. In each database, all search strategies will consider records from inception up to July 31, 2022. Additional sources will be also searched, including clinicaltrials.gov-NIH, the British Library, Google Scholar, and preprints for Health Sciences [medRXiv]. This systematic review will have no restrictions on languages or the settings of the target population. Additionally, we will scrutinize the reference lists of articles to search for additional studies [22]. The PECO acronym [23], that is,
P-Population = Adult patients [≥ 18 years of age] diagnosed with malignant neoplasm;
E-Exposure = Chemotherapy treatment; C-Comparison = not applicable;
O-Outcomes = Prevalence, composition, stability, and severity, as well as predictors of cancer symptom clusters, was used to answer our research question (what scientific evidence is available on the prevalence, composition, severity, and predictors of cancer symptom clusters in adult cancer patients undergoing chemotherapy?).

EndNote™ will be used to store, organize, and manage all retrieved studies. Study selection will be conducted by two independent reviewers (LCLJ and TF) using the Rayyan™ application. Controlled descriptors, such as MeSH terms, Emtree terms, Thesaurus, Cinahl headings, DeCS, and their synonyms will be screened. Keywords will be also identified. The Boolean operators "AND," "OR" and "NOT" will be employed to combine the descriptors [24, 25]. The preliminary pilot search strategy combining MeSH terms, synonyms, and keywords used in MEDLINE/PubMed is detailed in Table 1.

## Eligibility criteria

All observational and experimental study designs will be included.

**Population.**

- Inclusion criteria: Young adults and adults of both sexes, age > 18 years, and of any ethnicity.

- Exclusion criteria: Children, adolescents, pregnant women, and elderly people with cancer of both sexes. Age in this study is defined according to the MeSH term "Aged": a person aged 65–79 years or more.

**Intervention/Exposure.**

- Inclusion criteria: Adult cancer patients undergoing chemotherapy treatment.

- Exclusion criteria: Adult cancer patients undergoing radiotherapy or oncologic surgery.

**Outcomes.**

- Inclusion criteria: Prevalence, composition, severity, and predictors of cancer symptom clusters in young people and adults with cancer (> 18 years) resulting from chemotherapy treatment.

**Table 1. Preliminary pilot search strategy in MEDLLINE/PubMed.**

| Database | Search strategy |
| --- | --- |
| MEDLINE/ PubMed | • **Population** |
| | **#1**(("Young Adult" [MeSH Terms] OR "Adult" [MeSH Terms])) |
| | **#2** (("Neoplasms" [MeSH Terms] OR "Neoplasia" [All Fields] OR "Neoplasias" [All Fields] OR "Neoplasm" [All Fields] OR "Tumors" [All Fields] OR "Tumor" [All Fields] OR "Cancer" [All Fields] OR "Cancers" [All Fields] OR "Malignancy" [All Fields] OR "Malignancies" [All Fields] OR "Malignant Neoplasms" [All Fields] OR "Malignant Neoplasm" [All Fields] OR "Neoplasm, Malignant" [All Fields] OR "Neoplasms, Malignant" [All Fields])) |
| | **# 3** #1 AND #2 |
| | • **Exposure** |
| | **#4** (("Chemotherapy" [All Fields] OR "Chemotherapy, Adjuvant" [MeSH Terms] OR "Induction Chemotherapy" [MeSH Terms] OR "Consolidation Chemotherapy" [MeSH Terms] OR "Maintenance Chemotherapy" [MeSH Terms] NOT "Radiotherapy" [MeSH Terms] NOT "Surgery" [All Fields])) |
| | • **Outcomes** |
| | **#5**(("Symptom Cluster" [All Fields] OR "Cluster, Symptom" [All Fields] OR "Clusters, Symptom" [All Fields] OR "Symptom Clusters" [All Fields] OR "Cancer Symptom Clusters" [All Fields] OR "Symptom Constellation" [All Fields] OR "Symptom Management" [All Fields])) |
| | **#6** #3 AND #4 AND #5 |

- Exclusion criteria: Studies reporting the prevalence and severity of cancer symptom clusters in young people and adults who had undergone radiotherapy or oncologic surgery.

### Studies.

- Inclusion criteria: Observational and experimental studies.

- Exclusion criteria: Qualitative studies, guidelines and reviews.

The reference lists will be searched to seek additional studies. No restrictions regarding language, period of publication, or settings will be employed.

## Study selection

Initially, all records scrutinized from the eight electronic databases will be imported into End-Note™. Thus, duplicate studies will be removed. Two independent researchers (LCLJ and TF) will be search and screen the records by titles and abstracts using the Rayyan™ app. Following the initial screening, the full text of the retrieved studies will be assessed for inclusion/exclusion by two independent reviewers using the Rayyan™ app. Disagreements in the selected studies will be resolved by a third reviewer (KZN). A flowchart summarizing the study selection process in line with the PRISMA 2020 statement [26] is presented in Fig 1.

## Data extraction and synthesis

Data extraction will be performed by two reviewers (LCLJ and TF) for each included study, based on previously published forms [22, 24, 25, 27–29]. The same two reviewers (LCLJ and

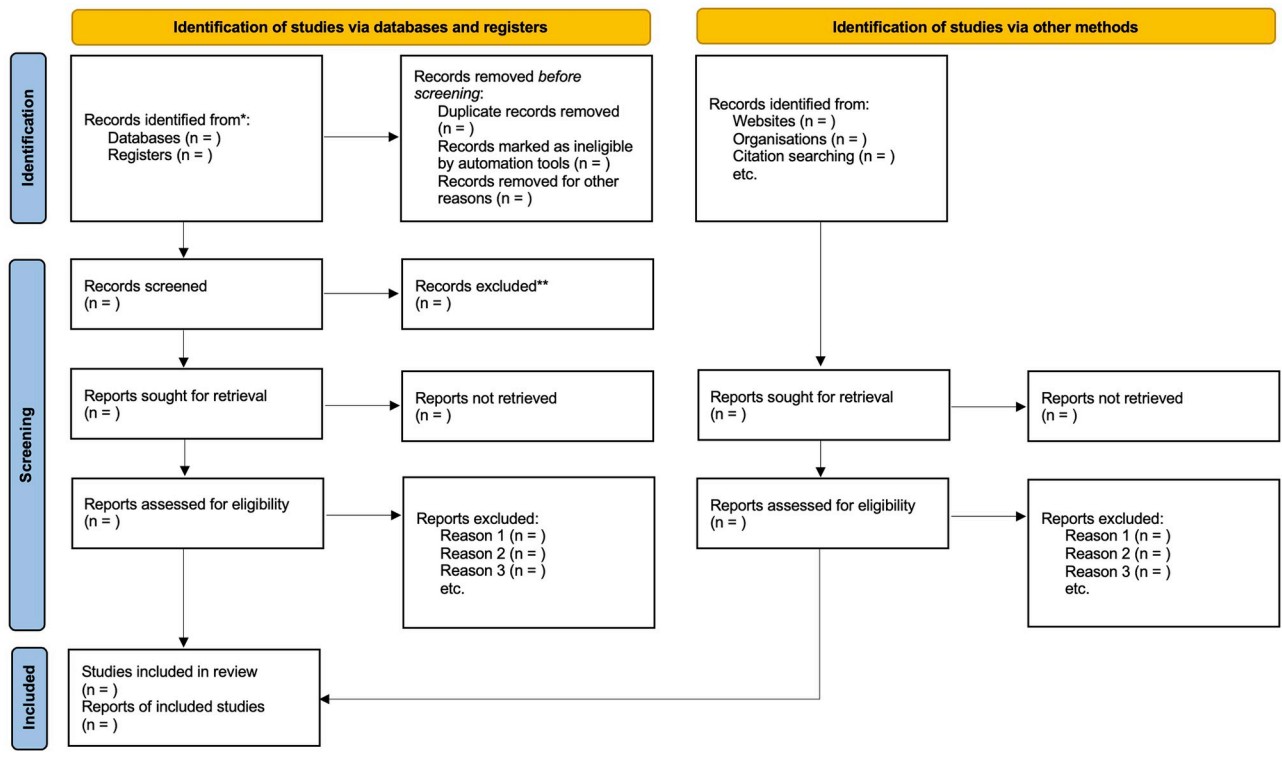

**Fig 1. PRISMA flowchart [26].**

TF) will independently perform the data extraction. The information to be extracted is detailed in Table 2.

### Critical appraisal of the included studies

Initially, the level of evidence will be identified and classified according to the scale developed by the Oxford Center for Evidence-Based Medicine, which is based on the research design and classifies them as 1A, 1B, 1C, 2A, 2B, 2C, 3A, 3B, 4, and 5 [30]. The internal validity and risk of bias of randomized controlled trials will be assessed using the revised Cochrane risk-of-bias tool for randomized trials (RoB 2) [31]. In addition, to assess nonrandomized controlled trials, the risk of bias in nonrandomized studies of interventions (ROBINS-I) will be used [32]. The Newcastle-Ottawa Scale (NOS) [33] will be used to evaluate the internal validity and risk of bias of the cohort studies. The Critical Appraisal Skills Programme (CASP) tool [34] will be employed for case-control studies. Cross-sectional studies will be assessed using the Agency for Healthcare Research and Quality (AHRQ) tool [35]. The same two reviewers (LCLJ and TF) will independently perform the critical appraisal.

### Data synthesis

The study characteristics will be summarized and presented in tables. Heterogeneity among studies will be measured using the $I^2$ statistic to estimate the percentage of variation across studies, ranging from 0% to 100% [36, 37], and its interpretation is as follows: a) $I^2$ = 0%–40%, low heterogeneity; b) $I^2$ = 30%–60%, moderate heterogeneity; c) $I^2$ = 50%–90%, substantial heterogeneity; and d) $I^2$ = 75%–100%, high heterogeneity [37, 38]. Based on the $I^2$ statistic, we will determine whether a meta-analysis is feasible [28, 29, 39].

Moreover, subgroup analysis will be performed using a random-effect model adjusted for age, sex, cancer symptom cluster prevalence, and cancer type. For data analysis, the pooled effect estimates will be calculated considering a 95% confidence interval (CI) and alpha set at 0.05, using R statistical software v. 4.0.4 (R Foundation for Statistical Computing, Vienna, Austria). In addition, the certainty of evidence will be rated based on Cochrane methods and in accordance with the Grading of Recommendations, Assessment, Development, and Evaluation (GRADE) [40]. Two reviewers (LCLJ and LBCP) will independently perform the critical appraisal of the evidence to assess the studies. Disagreements will be resolved by a third reviewer (KZN).

### Ethics aspects and plans for dissemination

Ethical approval was not required for this study design. Moreover, this systematic review and meta-analysis will be conducted following the PRISMA 2020 statement [26]. Regarding plans for dissemination, we intend to disclose the results via peer-reviewed publications and presentations at international conferences.

### Discussion

Indeed, there is a high prevalence of cancer symptom clusters in adult cancer patients, and such unpleasant symptoms are reported daily by patients and health professionals [4, 9, 16] in clinical practice. These symptom clusters are associated with worse prognosis and decreased quality of life. Thus, its effective management is crucial in oncology. The importance of evaluating and intervening in cancer symptom clusters, not only in terms of patient survival rate but also in terms of quality of life across the treatment, is a priority and an integral part of the

**Table 2. The data extraction form was based on previous publications [22, 24, 25, 27–29].**

| Study number: | Level of evidence: |
|---|---|
| | Methodological appraisal tool: |

| STUDY CARACTERISTICS | |
|---|---|
| Authors | |
| Title | |
| Year of publication | |
| Country | |
| Conflicts of interests | |
| Sponsorship | |
| Background | |
| Rationale | |
| Hypothesis | |
| Objectives | |

| Methods | |
|---|---|
| Methodology is reported is in compliance with STROBE (observational studies) or CONSORT (clinical trials) | |
| (   ) Yes | |
| (   ) No | |
| (   ) Partially | |
| Study design | |
| Local: | |
| Sample size: | |
| Inclusion criteria (definition of exposure of interest) | |
| Exclusion criteria | |
| Confounding factors/Interaction factors considered | |
| Ethical aspects | |
| Procedure for data collection: | |
| Instruments for data collection | |
| Outcomes / Evaluation of outcomes (Prevalence /composition/stability/severity/ predictors of cancer symptom clusters) | |
| Follow-up | |
| Statistical analysis | |
| If, cohort study | I. Number of participants in the exposed and unexposed cohort: |
| | II. Number of participants in each group: |
| | III. Comparability of exposed and unexposed cohorts |
| | IV. Contamination (unexposed patient being exposed): |
| | V. Follow-up period: |
| | VI. Dropouts: |
| If, case-control study | I. Criteria for selection of cases: |
| | II. Criteria for selection of controls: |
| | III. Comparability of groups: |
| | IV. Dropouts: |
| If, experimental or quase-experimental study | **a)** Trial Register: |
| | **b)** Trial arms: |
| | • Experimental Group: |
| | **c)** Randomization: |
| | **d)** Masking: |
| | **e)** Intervention protocol: |
| | **f)** Per-protocol and modified intention-to-treat analyses: |
| | • Per-protocol: |
| | • Intention-to-treat: |
| | • Dropouts: |

*(Continued)*

**Table 2.** (Continued)

| Study number: | Level of evidence: | | |
| --- | --- | --- | --- |
| | Methodological appraisal tool: | | |
| *Results* | | | |
| Main results | | | |
| Clinical significance | | | |
| Limitations of the study | | | |
| Strengths of the study | | | |
| *Conclusions* | | | |
| Main conclusions | | | |
| Implication for clinical practice and research or for decision-makers / stakeholders | | | |

pillars of research in oncology and for the advancement of science in symptom management in oncology [41–45].

## Conclusion

To the best of our knowledge, this systematic review will be the first to identify and critically assess evidence of the prevalence, composition, severity, and predictors of cancer symptom clusters in adult cancer patients undergoing chemotherapy. We intend to provide health professionals with subsidies to reflect on a better understanding of symptom clusters in adult cancer patients undergoing chemotherapy, with the aim of contributing to the development of evidence-based therapeutic strategies and success in clinical practice.

## Supporting information

**S1 Checklist. PRISMA checklist.**
(DOCX)

## Author Contributions

**Conceptualization:** Luís Carlos Lopes-Júnior, Taiani Ferrarini.

**Data curation:** Luís Carlos Lopes-Júnior, Luciana Bicalho Cevolani Pires, Jonathan Grassi Rodrigues, Luciane Bresciani Salaroli, Karolini Zuqui Nunes.

**Formal analysis:** Luís Carlos Lopes-Júnior, Luciana Bicalho Cevolani Pires, Jonathan Grassi Rodrigues, Luciane Bresciani Salaroli, Karolini Zuqui Nunes.

**Investigation:** Luís Carlos Lopes-Júnior.

**Methodology:** Luís Carlos Lopes-Júnior.

**Project administration:** Luís Carlos Lopes-Júnior.

**Software:** Luís Carlos Lopes-Júnior.

**Supervision:** Luís Carlos Lopes-Júnior.

**Validation:** Luís Carlos Lopes-Júnior, Luciana Bicalho Cevolani Pires, Jonathan Grassi Rodrigues, Karolini Zuqui Nunes.

**Visualization:** Luís Carlos Lopes-Júnior, Taiani Ferrarini, Luciana Bicalho Cevolani Pires, Jonathan Grassi Rodrigues, Luciane Bresciani Salaroli, Karolini Zuqui Nunes.

**Writing – original draft:** Luís Carlos Lopes-Júnior, Taiani Ferrarini, Luciana Bicalho Cevolani Pires, Jonathan Grassi Rodrigues, Karolini Zuqui Nunes.

**Writing – review & editing:** Luís Carlos Lopes-Júnior, Luciana Bicalho Cevolani Pires, Jonathan Grassi Rodrigues, Luciane Bresciani Salaroli, Karolini Zuqui Nunes.

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
