## [Decision Letter · Decision Letter 0]

6 May 2022

PONE-D-21-25793Cancer symptom clusters in adult patients undergoing chemotherapy: a systematic review protocolPLOS ONE

Dear Dr. Lopes-Junior,

Thank you for submitting your manuscript to PLOS ONE. After careful consideration, we feel that it has merit but does not fully meet PLOS ONE’s publication criteria as it currently stands. Therefore, we invite you to submit a revised version of the manuscript that addresses the points raised during the review process.

We have now received reports from our referees of your manuscript, as agree with reviewers comments raised a few concerns about this study. After careful consideration, we invite you to submit a revised version of the manuscript.  

We look forward to receiving your revised manuscript.

Kind regards,

Senthilnathan Palaniyandi, Ph.D

Academic Editor

PLOS ONE

Reviewers' comments:

Reviewer's Responses to Questions

**Comments to the Author**

1. Does the manuscript provide a valid rationale for the proposed study, with clearly identified and justified research questions?

Reviewer #1: Yes

Reviewer #2: No

Reviewer #3: Yes

2. Is the protocol technically sound and planned in a manner that will lead to a meaningful outcome and allow testing the stated hypotheses?

Reviewer #1: Yes

Reviewer #2: Partly

Reviewer #3: Yes

3. Is the methodology feasible and described in sufficient detail to allow the work to be replicable?

Reviewer #1: Yes

Reviewer #2: Yes

Reviewer #3: Yes

4. Have the authors described where all data underlying the findings will be made available when the study is complete?

Reviewer #1: Yes

Reviewer #2: Yes

Reviewer #3: Yes

5. Is the manuscript presented in an intelligible fashion and written in standard English?

Reviewer #1: Yes

Reviewer #2: No

Reviewer #3: Yes

6. Review Comments to the Author

You may also provide optional suggestions and comments to authors that they might find helpful in planning their study.

Reviewer #1: I appreciate the opportunity to review this article protocol. This  systematic review is important in clinical practice, especially in  the developing countries that have high mortality rates in cancer patients.There are currently no comments on the study protocol, but there is the systematic review (Skye Tian Dong, et al 2014 "Symptom Clusters in Patients With Advanced Cancer: A Systematic Review of  Observational Studies"), I hope to return to. I wish the researchers success and await such the important  systematic review. No need to chart1

Reviewer #2: The authors aim to identify and gather evidence on the prevalence, composition, severity, and predictors of cancer symptom clusters in adult cancer patients undergoing chemotherapy. However, a similar study has recently been carried out elsewhere and the results have been published: Harris CS, Kober KM, Conley YP, et alSymptom clusters in patients receiving chemotherapy: A systematic review; BMJ Supportive & Palliative Care 2022;12:10-21. Hence, this study is not warranted.

Reviewer #3: The present is a protocol aiming to capture clusters of symptoms of patients undergoing chemiotherapy. The paper is very clear regarding goal source of participants and outcomes

Some issues are needed

1) Abstract and full paper>why will search be stopped on August 2021?

2) Introduction and methods>authors spoke correctly about physical and physiological symptoms. Do they think subgroup analysis may be needed?

3) Methods: elderly patients should be defined with a clear cut off or reported according to definition of each papers

4) Methods: regarding features of the studies I’m not sure that impact factor may be so relevant (just a comments) as IF does not always relate with quality of the paper

5) Conflict of interests>please define better

6) Methods>do authors think that a subgroup analysis for kinf of cancer and for protocol of chemiptherapy may be useful?

7) Moreover, do they think that some quantitative analysis (e.g meta-regression to evaluate impact of gender, age,) on clusters may be of interest?

8) Methods: primary and secondary end points should be better defined

7. PLOS authors have the option to publish the peer review history of their article (what does this mean?). If published, this will include your full peer review and any attached files.

Reviewer #1: No

Reviewer #2: No

Reviewer #3: **Yes: **Fabrizio D'Ascenzo

---

## [Author Response · Author response to Decision Letter 0]

17 May 2022

Vitoria,ES, May 15, 2022

Dear Dr. Senthilnathan Palaniyandi,

We would like to thank you for the opportunity to review the manuscript after the reviewers' suggestions and recommendations.

All points were addressed and/or clarified in this new version. In addition, we responded item by item to the questions raised by the reviewers in this letter.

Comments to the Author

1. Does the manuscript provide a valid rationale for the proposed study, with clearly identified and justified research questions?

Reviewer #1: Yes

Reviewer #2: No

Reviewer #3: Yes

2. Is the protocol technically sound and planned in a manner that will lead to a meaningful outcome and allow testing the stated hypotheses?

Reviewer #1: Yes

Reviewer #2: Partly

Reviewer #3: Yes

3. Is the methodology feasible and described in sufficient detail to allow the work to be replicable?

Reviewer #1: Yes

Reviewer #2: Yes

Reviewer #3: Yes

4. Have the authors described where all data underlying the findings will be made available when the study is complete?

Reviewer #1: Yes

Reviewer #2: Yes

Reviewer #3: Yes

5. Is the manuscript presented in an intelligible fashion and written in standard English?

Reviewer #1: Yes

Reviewer #2: No

Reviewer #3: Yes

6. Review Comments to the Author

Reviewer #1: I appreciate the opportunity to review this article protocol. This systematic review is important in clinical practice, especially in the developing countries that have high mortality rates in cancer patients.There are currently no comments on the study protocol, but there is the systematic review (Skye Tian Dong, et al 2014 "Symptom Clusters in Patients With Advanced Cancer: A Systematic Review of Observational Studies"), I hope to return to. I wish the researchers success and await such the important systematic review. No need to chart1

Response: Thank you so much for your comments! We have added a paragraph mentioning this review (Dong et al., 2014) in the introduction and specified how ours differs from this one as well as the current one by Harris et al., 2022. In addition, we have removed the chart 1 as per suggested. Thanks!

Reviewer #2: The authors aim to identify and gather evidence on the prevalence, composition, severity, and predictors of cancer symptom clusters in adult cancer patients undergoing chemotherapy. However, a similar study has recently been carried out elsewhere and the results have been published: Harris CS, Kober KM, Conley YP, et alSymptom clusters in patients receiving chemotherapy: A systematic review; BMJ Supportive & Palliative Care 2022;12:10-21. Hence, this study is not warranted.

Response: We have added a paragraph mentioning this review (Dong et al., 2014) in the introduction and specified how ours differs from this one.

“Recently, a systematic review was published with the objective of evaluating the progress in symptom clusters research in adults receiving primary or adjuvant chemotherapy since 2016, and showed that psychological, gastrointestinal, and nutritional clusters were the most commonly identified clusters. Only the psychological clusters remained relatively stable over time [19].

It is stand out that our review study differs from the latter in the following aspects: we will not specify a publication date or language limit (in order to minimize publication bias). Furthermore, we will explore beyond observational studies, as we will include experimental studies and also, we will expand the number of databases to 8, besides the access gray literature and pre-prints for Health Sciences. Another difference is that both previous reviews used only generic tools for methodological appraisal and that assess the report of the study more than the risk of bias (in terms of internal validity and external validity of the studies). Here, we will use valid and design-specific tools, following the recommendations of the Cochrane Collaboration [20]. Additionally, there is still no systematic review and metanalysis taking into account a robust assessment of the risk of bias in studies involving cancer symptom clusters with validated and design-specific tools, which justifies our study for a potential contribution to the area”.

Reviewer #3: The present is a protocol aiming to capture clusters of symptoms of patients undergoing chemiotherapy. The paper is very clear regarding goal source of participants and outcomes

Response: Thank you so much!

Some issues are needed

1) Abstract and full paper>why will search be stopped on August 2021?

Response: In fact, we put August, 2021 because that was the date when we submitted the article to PLOS ONE. However, we will update the search once the protocol is accepted. We edit this in the paper. Thank you!

2) Introduction and methods>authors spoke correctly about physical and physiological symptoms. Do they think subgroup analysis may be needed?

Response: This is an important point to consider. Thank you very much! Yes, probably depending on the results we will find we will do subgroup analysis. We really appreciated this comment. I have added one paragraph on the sub-item “Data synthesis” into the methods section.

3) Methods: elderly patients should be defined with a clear cut off or reported according to definition of each papers

Response: Thanks for this comment. We have added one sentence in order to make it clearer. “Aged in this study is defined according to MeSH term “Aged”: a person 65 through 79 years of age or more”.

Aged

A person 65 through 79 years of age. For a person older than 79 years, AGED, 80 AND OVER is available.

Year introduced: 1966

4) Methods: regarding features of the studies I’m not sure that impact factor may be so relevant (just a comments) as IF does not always relate with quality of the paper

Response: Indeed, we agreed with you. So, we have removed. Thanks!

5) Conflict of interests>please define better

Response: The Committee on Publication Ethics (COPE) states in its Guidelines on Good Publication Practice (2003) that: 'Conflicts of interest arise when authors, reviewers, or editors have interests that are not fully apparent and that may influence their judgments on what is published.

6) Methods>do authors think that a subgroup analysis for kind of cancer and for protocol of chemiptherapy may be useful?

Response: This point is very interesting. Studies with cancer symptom clusters are quite heterogeneous around the target population, types of cancers included in the analyses, tumor staging and chemotherapy protocol. Thus, this stratification can be carried out depending on the evidence that we will find, especially if the meta-analysis is feasible. I have added this on “Data Synthesis”. Many Thanks!

7) Moreover, do they think that some quantitative analysis (e.g meta-regression to evaluate impact of gender, age,) on clusters may be of interest?

Response: Yes. I specified this in the topic “Data Syjthesis”. Thank you so much for your valuable contribuitions.

8) Methods: primary and secondary end points should be better defined

Response: We have edited this in the chart 1. Actually, our outcomes are including: “Prevalence /composition/stability/ severity/ predictors of cancer symptom clusters, as stated in the PECO acronym.

7. PLOS authors have the option to publish the peer review history of their article (what does this mean?). If published, this will include your full peer review and any attached files.

Do you want your identity to be public for this peer review? For information about this choice, including consent withdrawal, please see our Privacy Policy.

Reviewer #1: No

Reviewer #2: No

Reviewer #3: Yes: Fabrizio D'Ascenzo

---

## [Decision Letter · Decision Letter 1]

12 Jul 2022

PONE-D-21-25793R1Cancer symptom clusters in adult patients undergoing chemotherapy: a systematic review and metanalysis protocolPLOS ONE

Dear Dr. Lopes-Júnior,

Thank you for submitting your manuscript to PLOS ONE. After careful consideration, we feel that it has merit but does not fully meet PLOS ONE’s publication criteria as it currently stands. Therefore, we invite you to submit a revised version of the manuscript that addresses the points raised during the review process. Your manuscript has been reassessed by the three reviewers from the previous round, whose reports can be found below. As you will see from the comments, the reviewers acknowledge that the manuscript has improved significantly. While assessing your manuscript, we noticed several errors in the use of spelling, punctuation and grammar throughout the manuscript. So that your manuscript meets our publication criterion requiring that 'the article is presented in an intelligible fashion and is written in standard English.', we request that you thoroughly copyedit your manuscript to fix these errors. You may wish to use a professional scientific editing service to do this. 

We look forward to receiving your revised manuscript.

Kind regards,

Joseph Donlan

Editorial Office

PLOS ONE

Journal Requirements:

Reviewers' comments:

Reviewer's Responses to Questions

**Comments to the Author**

1. Does the manuscript provide a valid rationale for the proposed study, with clearly identified and justified research questions?

Reviewer #1: Yes

Reviewer #2: Partly

Reviewer #3: Yes

2. Is the protocol technically sound and planned in a manner that will lead to a meaningful outcome and allow testing the stated hypotheses?

Reviewer #1: Yes

Reviewer #2: Partly

Reviewer #3: Yes

3. Is the methodology feasible and described in sufficient detail to allow the work to be replicable?

Reviewer #1: Yes

Reviewer #2: Yes

Reviewer #3: Yes

4. Have the authors described where all data underlying the findings will be made available when the study is complete?

Reviewer #1: Yes

Reviewer #2: Yes

Reviewer #3: Yes

5. Is the manuscript presented in an intelligible fashion and written in standard English?

Reviewer #1: Yes

Reviewer #2: No

Reviewer #3: Yes

6. Review Comments to the Author

You may also provide optional suggestions and comments to authors that they might find helpful in planning their study.

Reviewer #1: When mentioning this review(...2014), this review was not intended in itself, but rather the intent was, the authors should clarify the strengths of their study and its difference from what preceded it, especially since the authors indicated in the study protocol that (this systematic review and comprehensive analysis will be the first to identify and evaluate Evidence about prevalence, composition, and severity ....)

I wish you success in publication a valuable study.

Reviewer #2: Thank you for addressing the earlier comments. The manuscript needs a thorough proof reading for syntactic errors and grammars, especially in the paragraphs that have been added now to address the reviewers' comments.

Reviewer #3: all comments have been addressed.

i have no further comments, because the authors cleary detailed all the comments

7. PLOS authors have the option to publish the peer review history of their article (what does this mean?). If published, this will include your full peer review and any attached files.

Reviewer #1: **Yes: **Rania E Moustafa

Reviewer #2: No

Reviewer #3: **Yes: **Fabrizio D'Ascenzo

---

## [Author Response · Author response to Decision Letter 1]

15 Jul 2022

Response to Reviewers

Vitória, ES, Brazil, July 15th, 2022

Minor revision

PONE-D-21-25793R1

Cancer symptom clusters in adult patients undergoing chemotherapy: a systematic review and metanalysis protocol

Dear Dr. Lopes-Júnior,

Thank you for submitting your manuscript to PLOS ONE. After careful consideration, we feel that it has merit but does not fully meet PLOS ONE’s publication criteria as it currently stands. Therefore, we invite you to submit a revised version of the manuscript that addresses the points raised during the review process.

Your manuscript has been reassessed by the three reviewers from the previous round, whose reports can be found below. As you will see from the comments, the reviewers acknowledge that the manuscript has improved significantly.

Response: Dear Dr. Joseph Donlan,

Thank you for the opportunity to review the manuscript after the reviewers' suggestions and recommendations.

All points were addressed and/or clarified in this new version. In addition, we responded item by item to the questions raised by the reviewers in this letter.

While assessing your manuscript, we noticed several errors in the use of spelling, punctuation and grammar throughout the manuscript. So that your manuscript meets our publication criterion requiring that 'the article is presented in an intelligible fashion and is written in standard English.', we request that you thoroughly copyedit your manuscript to fix these errors. You may wish to use a professional scientific editing service to do this. 

Response: Ok. As suggested, we have done it. So, we have paid an English proofreading to Editage (https://www.editage.com) for polishing the text and correct grammar and spelling mistakes according to the cultured language. Therefore, an extensive review of English was carried out. 

We look forward to receiving your revised manuscript.

Kind regards,

Joseph Donlan

Editorial Office

PLOS ONE

Journal Requirements:

Response: We have carefully reviewed the reference list and we ensure that it is complete and correct. Also, we have no papers retracted cited in this manuscript.

Review Comments to the Author

Reviewer #1: When mentioning this review(...2014), this review was not intended in itself, but rather the intent was, the authors should clarify the strengths of their study and its difference from what preceded it, especially since the authors indicated in the study protocol that (this systematic review and comprehensive analysis will be the first to identify and evaluate Evidence about prevalence, composition, and severity ....)

I wish you success in publication a valuable study.

Response: OK. We addressed this in the introduction. Thank you so much!

Reviewer #2: Thank you for addressing the earlier comments. The manuscript needs a thorough proof reading for syntactic errors and grammars, especially in the paragraphs that have been added now to address the reviewers' comments.

Response: Thank you so much! We have paid an English proofreading to Editage (https://www.editage.com) for polishing the text and correct grammar and spelling mistakes according to the cultured language. Therefore, an extensive review of English was carried out.

Reviewer #3: all comments have been addressed.

i have no further comments, because the authors cleary detailed all the comments

Response: Thank you so much!

Reviewer #1: Yes: Rania E Moustafa

Reviewer #2: No

Reviewer #3: Yes: Fabrizio D'Ascenzo

Yours Sincerely, 

The authors,

---

## [Decision Letter · Decision Letter 2]

9 Aug 2022

Cancer symptom clusters in adult patients undergoing chemotherapy: a systematic review and metanalysis protocol

PONE-D-21-25793R2

Dear Dr. Lopes-Júnior,

We’re pleased to inform you that your manuscript has been judged scientifically suitable for publication and will be formally accepted for publication once it meets all outstanding technical requirements.

Kind regards,

James Mockridge

Staff Editor

PLOS ONE

Reviewers' comments:

Reviewer's Responses to Questions

**Comments to the Author**

1. Does the manuscript provide a valid rationale for the proposed study, with clearly identified and justified research questions?

Reviewer #1: Yes

Reviewer #3: Yes

2. Is the protocol technically sound and planned in a manner that will lead to a meaningful outcome and allow testing the stated hypotheses?

Reviewer #1: Yes

Reviewer #3: Yes

3. Is the methodology feasible and described in sufficient detail to allow the work to be replicable?

Reviewer #1: Yes

Reviewer #3: Yes

4. Have the authors described where all data underlying the findings will be made available when the study is complete?

Reviewer #1: Yes

Reviewer #3: Yes

5. Is the manuscript presented in an intelligible fashion and written in standard English?

Reviewer #1: Yes

Reviewer #3: Yes

6. Review Comments to the Author

You may also provide optional suggestions and comments to authors that they might find helpful in planning their study.

Reviewer #1: Thanks for reviewing this article.

for There are no comments. Because the authors made all points clear.

Reviewer #3: All comments have been addressed. Authors should be complimented for performing such an accurate analysis

7. PLOS authors have the option to publish the peer review history of their article (what does this mean?). If published, this will include your full peer review and any attached files.

Reviewer #1: **Yes: **Rania E Moustafa

Reviewer #3: **Yes: **Fabrizio D'Ascenzo

---

## [Editor Report · Acceptance letter]

23 Aug 2022

PONE-D-21-25793R2 

Cancer symptom clusters in adult patients undergoing chemotherapy: a systematic review and meta-analysis protocol 

Dear Dr. Lopes-Júnior:

I'm pleased to inform you that your manuscript has been deemed suitable for publication in PLOS ONE. Congratulations! Your manuscript is now with our production department. 

Kind regards, 

on behalf of

Dr Joseph Donlan 

Staff Editor

PLOS ONE